# Clinical Impact of Functional CYP2C19 and CYP2D6 Gene Variants on Treatment with Antidepressants in Young People with Depression: A Danish Cohort Study

**DOI:** 10.3390/ph15070870

**Published:** 2022-07-14

**Authors:** Liv S. Thiele, Kazi Ishtiak-Ahmed, Janne P. Thirstrup, Esben Agerbo, Carin A. T. C. Lunenburg, Daniel J. Müller, Christiane Gasse

**Affiliations:** 1Department of Affective Disorders, Aarhus University Hospital Psychiatry, 8200 Aarhus, Denmark; 201409442@post.au.dk (L.S.T.); kazahm@rm.dk (K.I.-A.); janne.thirstrup@biomed.au.dk (J.P.T.); lunenburg.c@gmail.com (C.A.T.C.L.); 2Department of Clinical Medicine, Aarhus University, 8200 Aarhus, Denmark; 3Department of Biomedicine, Aarhus University, 8000 Aarhus, Denmark; 4National Centre for Register-Based Research (NCRR), Aarhus BSS, Aarhus University, 8210 Aarhus, Denmark; ea@econ.au.dk; 5Centre for Integrated Register-Based Research Aarhus University (CIRRAU), 8210 Aarhus, Denmark; 6Pharmacogenetics Research Clinic, Campbell Family Mental Health Research Institute, Centre for Addiction and Mental Health, Toronto, ON M6J 1H4, Canada; daniel.mueller@camh.ca; 7Department of Psychiatry, University of Toronto, Toronto, ON M5S 1A1, Canada; 8Psychosis Research Unit, Aarhus University Hospital Psychiatry, 8200 Aarhus, Denmark

**Keywords:** pharmacogenetics, antidepressants, utility, population-based

## Abstract

Background: The clinical impact of the functional CYP2C19 and CYP2D6 gene variants on antidepressant treatment in people with depression is not well studied. Here, we evaluate the utility of pharmacogenetic (PGx) testing in psychiatry by investigating the association between the phenotype status of the cytochrome P450 (CYP) 2C19/2D6 enzymes and the one-year risks of clinical outcomes in patients with depression with incident new-use of (es)citalopram, sertraline, or fluoxetine. Methods: This study is a population-based cohort study of 17,297 individuals who were born between 1981 and 2005 with a depression diagnosis between 1996 and 2012. Using array-based single-nucleotide-polymorphism genotype data, the individuals were categorized according to their metabolizing status of CYP2C19/CYP2D6 as normal (NM, reference group), ultra-rapid- (UM), rapid- (RM), intermediate- (IM), or poor-metabolizer (PM). The outcomes were treatment switching or discontinuation, psychiatric emergency department contacts, and suicide attempt/self-harm. By using Poisson regression analyses, we have estimated the incidence rate ratios (IRR) with 95% confidence intervals (95% CI) that were adjusted for covariates and potential confounders, by age groups (<18 (children and adolescents), 19–25 (young adults), and 26+ years (adults)), comparing the outcomes in individuals with NM status (reference) versus the mutant metabolizer status. For statistically significant outcomes, we have calculated the number needed to treat (NNT) and the number needed to genotype (NNG) in order to prevent one outcome. Results: The children and adolescents who were using (es)citalopram with CYP2C19 PM status had increased risks of switching (IRR = 1.64 [95% CI: 1.10–2.43]) and suicide attempt/self-harm (IRR = 2.67 [95% CI; 1.57–4.52]). The young adults with CYP2C19 PM status who were using sertraline had an increased risk of switching (IRR = 2.06 [95% CI; 1.03–4.11]). The young adults with CYP2D6 PM status who were using fluoxetine had an increased risk of emergency department contacts (IRR = 3.28 [95% CI; 1.11–9.63]). No significant associations were detected in the adults. The NNG for preventing one suicide attempt/suicide in the children who were using (es)citalopram was 463, and the NNT was 11. Conclusion: The CYP2C19 and CYP2D6 PM phenotype statuses were associated with outcomes in children, adolescents, and young adults with depression with incident new-use of (es)citalopram, sertraline, or fluoxetine, therefore indicating the utility of PGx testing, particularly in younger people, for PGx-guided antidepressant treatment.

## 1. Introduction

Antidepressants are essential of the pharmacological treatment of depression in youths and adults [1]. However, treatment with antidepressants is often not optimal, with about 30% of patients not recovering, even after several attempts of treatment with different antidepressants [2]. The insufficient treatment response and the adverse events may partly be attributed to the individual’s capacity to metabolize the antidepressant (pharmacokinetics), which is affected by genetic variations of drug-metabolizing enzymes, e.g., the hepatic cytochrome P450 (CYP) system [3]. In particular, the highly polymorphic enzymes CYP2C19 and CYP2D6 play a central role in the metabolism of many antidepressants, including the selective serotonin-reuptake inhibitors (SSRIs) (es)citalopram, sertraline, and fluoxetine [3]. There is an increasing body of clinical evidence linking pharmacogenetic (PGx) variability of CYP2D6 and CYP2C19 to drug blood concentrations [4,5], the treatment response [4,6], and the remission rates [6] in patients with depression. Thus, by PGx testing of the genotypes of CYP2C19 and CYP2D6 metabolizer phenotypes according to the variable genotypes’ activity can be classified into poor (PM), intermediate (IM), normal (NM), rapid (RM), or ultra-rapid metabolizer (UM) for the given enzyme. These phenotypes can guide the choice of drug and the dose adjustment in order to maximize the likelihood for treatment effectiveness and minimize the adverse events [7]. 

As a first-line treatment for depression, SSRIs are commonly used in the population worldwide, with more than 4% of the total Danish population using SSRIs in 2021 [8], 8% in UK in 2011 [9], and approximately 11% in the USA in 2021 [10]. The recommendations for PGx-guided dosing for PM and IM CYP2C19 phenotypes of the SSRIs (es)citalopram and sertraline have been published by the Dutch Pharmacogenetics Working Group (DPWG) and the Clinical Pharmacogenetics Implementation Consortium (CPIC) [11,12]. In addition, the drug labels of (es)citalopram and sertraline, by the national drug authorities in the USA, Switzerland, Canada, and Japan, consider PGx testing actionable, while the labelling for these drugs in the EU does not include any annotations for PGx testing [11]. Fluoxetine is the only approved SSRI for the treatment of depression in children in Denmark [13]. Fluoxetine is mainly metabolized by CYP2D6, but neither drug labels nor the DPWG or the CPIC offer dosing guidelines, due to insufficient relevant clinical evidence in children [11]. 

Despite the existing clinical evidence, drug labelling, actionable PGx recommendations for (es)citalopram and sertraline, and the frequent use of these first-line drugs in the population, the clinical utility of PGx testing is still broadly discussed nationally and internationally, particularly in youths [14], and the implementation of PGx testing remains low in psychiatry in Denmark [7,15], though it is increasing internationally [3]. The clinical utility of PGx can be defined as the ability of PGx-guided treatment and dosing to prevent the adverse effects expressed by the number needed to genotype (NNG) and number needed to treat (NNT) in order to avoid one adverse event [16]. Published PGx studies rarely report the clinical utility measures [17], which could support the communication of the evidence of PGx drugs with strong associations and/or frequent use in the population for the clinical implementation of PGx testing [16]. 

Here, we have examined the association of CYP2C19 and CYP2D6 gene variants translated into PGx phenotypes with treatment outcomes of switching or discontinuation, psychiatric emergency department contacts, and suicide attempt/self-harm in patients with depression with incident new-use of (es)citalopram, sertraline, or fluoxetine in children and adolescents (≤18 years), young adults (19–25), and adults (≥26 years). 

## 2. Results 

### 2.1. Characteristics of the Study Population 

Of the 24,110 individuals with a hospital depression diagnosis given at any time between 1 January 1996 and 31 December 2012, 20,343 (84%) had redeemed at least one prescription for (es)citalopram, sertraline, or fluoxetine. Of the latter, 17,297 (85%) had valid genetic data and, therefore, formed the study population, of which 70% were females, 90% were younger than 26 years, and 90% were of Danish/European origin. Compared to the excluded individuals, the individuals of the study population were slightly younger, had filled their first prescription for the respective antidepressants in the more recent years in the study period, had their first prescription issued from the hospital, and had a diagnosis of autism more often, and a diagnosis of schizophrenia or bipolar disorder less often (Appendix A).

Of the study population, the majority (62%) had redeemed at least one prescription for (es)citalopram during the study period (Table 1). According to the indicated use of fluoxetine in children, the mean age of the fluoxetine users was lower than in individuals initiating the other antidepressants. The differences in the baseline characteristics stratified by children and adolescents, young adults, and adults existed, but not regarding the frequency of CYP2C19 and CYP2D6 phenotypes (Appendix A and Table 2). 

### 2.2. Associations between the CYP2C19 and CYP2D6 Phenotypes and Clinical Outcomes

Overall, irrespective of the outcomes, the study population of 17,297 individuals contributed to a total follow-up time of 17,237 person-years (PYs) since the treatment initiation with the respective drugs, with a mean follow-up period of 364 days. During the study period, 793 individuals emigrated and 124 died. 

The incidence rates (IR) per 100 person-years with 95% CI of outcomes, according to the index drug use, are reported in Appendix A. Figure 1 and Appendix A describe the association between the CYP2C19 and CYP2D6 phenotypes and the clinical outcomes in the individuals with depression who were using (es)citalopram, sertraline, or fluoxetine by the age groups. 

The children and adolescents who were using (es)citalopram with a CYP2C19 PM status had an incident rate of switching of 41 per 100 PYs and a statistically significant increased risk of switching (IRR_PM_ = 1.64 [95% CI: 1.10–2.43]) compared to those with CYP2C19 NM status. The children and adolescents who were using (es)citalopram with a CYP2C19 PM status had an incident rate of 23 per 100 PYs of attempted suicide/self-harm and had a statistically significant increased risk of suicide attempt/self-harm (IRR_PM_ = 2.67 [95% CI; 1.57–4.52]), compared to those with CYP2C19 NM status (Figure 1a). 

Among the young adults who were using sertraline with CYP2C19 PM status, 51 per 100 PYs switched to another drug, with a statistically significant increased risk of switching (IRR_PM_ = 2.06 [95% CI; 1.03–4.11]) (Figure 1b) compared with CYP2C19 NMs. The young adults who were using fluoxetine with a CYP2D6 PM status had an IR of 55 psychiatric emergency department contacts per 100 PYs, with a more than three-fold increased risk of psychiatric emergency department contacts compared with CYP2D6 NMs (IRR_PM_ = 3.28 [95% CI; 1.11–9.63]) (Figure 1b). 

Among the adults, no statistically significant findings were detected, but associations indicating a U-shaped relationship across the phenotypes with higher risks in users of (es)citalopram with CYP2C19 PM and UM status were found (Figure 1c). 

### 2.3. Potential Clinical Validity and Population Impact of PGx Testing

Overall, the clinical utility and population impact of PGx testing for all of the statistically significant associations of switching, suicide attempt, and self-harm among children and adolescents, and young adults were 1–2.5% for PAF, the NNG was between 460 and 503, according to their metabolizer phenotypes, and the NNT was between 10 and 11 in order to prevent one outcome, (Table 3). 

## 3. Discussion 

We have studied the association between CYP2C19 and CYP2D6 gene variants and the treatment outcomes in children and adolescents, young adults, and adults with depression who were using (es)citalopram, sertraline, or fluoxetine. The associations were most pronounced in the children and adolescents with statistically significant results in PM of CYP2C19 who were using (es)citalopram, with regard to switching and suicide attempts/self-harm. We found U-shaped associations from UM to PM of CYP2C19 in both the children and adolescents, and the adults who were using (es)citalopram, related to suicide attempt/self-harm. The association measures that were translated into measures of clinical utility were nominally modest, which may be partly because CYP2C19 and CYP2D6 PM are rare phenotypes in a multifactorial setting of drug response. 

Compared with the previous findings of CYP2C19 genetic variability and switching, a study among adults [4] found a more than 3-fold increased frequency of switching within one year in CYP2C19 PM and RM/UM status who were using escitalopram compared with the none finding among PM in adults who were using (es)citalopram and the borderline non-significant association in the RM and UM adults in our study. In addition, we found a 64% significantly increased risk in the children and adolescents with CYP2C19 PM status who were using (es)citalopram. According to the authors, the increased risk of switching in CYP2C19 PMs and UM was explained by the increased (PM) and decreased (UM) drug-plasma concentrations [4], potentially leading to adverse events and an insufficient treatment response. Though pointing towards similar conclusions, our study differed from Jukic et al. in identifying the proxies for switching. The study by Jukic et al. was limited to data that was based on the therapeutic drug measurements, while our study was limited to prescription data. However, we were able to adjust for potential confounders and pheno-conversion, which may partly explain the differences in the sizes of the detected associations.

Regarding sertraline, we found that the young adults with PM status who were using sertraline were also more likely to switch. A systematic review and meta-analysis [5] showed that CYP2C19 PMs had higher sertraline plasma concentrations, while Poweleit et al. [18] showed that CYP2C19 status from PM to UM was inversely associated with sertraline doses at the beginning of treatment but not with doses in association with response. Overall, Jessel et al. [19] found that 10% of individuals tended to switch if they were using antidepressants that were not in line with their CYP2C19 and/or CYP2D6 status, compared with 6% of patients who were using antidepressants that were aligned with their metabolizer phenotype. 

Regarding discontinuation, Aldrich et al. [20] found a significant association between the discontinuation of (es)citalopram in youth with anxiety and/or depression and CYP2C19 PM and IM status, which is partly in line with our study, where the children and adolescents who were using (es)citalopram with a CYP2C19 IM phenotype had a slightly increased, but statistically insignificant, risk for discontinuation. By contrast, meta-analyses of clinical trials have reported that those with CYP2C19 PM status who were using escitalopram had an improved treatment response, higher rates of side effects, but had less drop out from the clinical trials [6,21].

Suicidal behavior is a feared and severe outcome in young patients using SSRIs [22]. We have detected a more than a 2-fold increased risk of suicide attempt/self-harm in the children and adolescents who were using (es)citalopram with poor metabolizing capacity. In the young adults and the adults of our study, CYP2C19 PMs who were using (es)citalopram also showed a nominal increased risk of suicide attempt/self-harm, as well as CYP2C19 UMs, but this was not statistically significant. It should be noted that our definition included self-harm of both known and unknown suicidal intent [23]. By contrast, an international study of 243 patients found no differences between the phenotypes and suicidal behavior, as measured by clinical rating scales [24], while a post-mortem study showed an enrichment of CYP2C19 PMs and UMs among adult suicide victims who had tested positively for citalopram, compared with the population controls [25]. It should be noted that the use of (es)citalopram dropped to nearly 0 by 2021 [26] in children and adolescents since the warnings of the increased risk of suicidal behavior were issued in 2010 [22]. In 2021, fluoxetine and sertraline were the most frequently used antidepressants among children in Denmark [26]. The numbers were too small to assess the suicidality in the users of fluoxetine, while a nominally increased risk was seen among the children and adolescents with PM status who were using sertraline. 

Regarding fluoxetine, according to previous reports, CYP2D6 metabolizer status showed no influence on 8- or 12-week fluoxetine treatment response, which was assessed with multiple disease severity scales in children and adolescents [27]. Here, we have found nominally decreased risks for all of the outcomes in the children and adolescents with CYP2D6 PM status, which may indicate a superior response to fluoxetine in CYP2D6 PMs, possibly due to higher drug-plasma concentrations in these patients [4], without the off-set of higher risks of adverse events leading to discontinuation or switching. In contrast to the children and adolescents, the young adults and the adults who were using fluoxetine with PM status in our study had an increased risk of switching and emergency room contacts, which is in line with a smaller study reporting that 33% of people with a CYP2D6 PM status discontinued the fluoxetine treatments compared with 14% of adults with a CYP2D6 NM status [28]. 

The metabolism of fluoxetine is complicated by the self-inhibition/pheno-conversion of CYP2D6 by fluoxetine enantiomers during chronic treatment, which increases the importance of alternative metabolic pathways, including CYP2C19 [13]. Thus, the CYP2C19 metabolism, and other alternative pathways, may compensate the limited CYP2D6 metabolism [29]. Due to the described complexity of the metabolic pathway, and the power issues regarding treatment outcomes, it would have been beyond the scope of the current study to evaluate the combinatorial effect of both the CYP2D6 and CYP2C19 variants, which should be addressed in future studies. 

### 3.1. Potential Clinical Validity and Population Impact of PGx Testing

Despite the significant associations, translated to the clinical utility measures, these appear to be quite modest. This is partly because CYP2C19 and CYP2D6 PM are rare phenotypes in a multifactorial setting of drug response. Yet, regarding the suicide attempts/self-harm in association with (es)citalopram, the NNG was 464 and the NNT was 10, indicating the utility of pre-emptive PGx testing in those patients for whom fluoxetine is not an option. Overall, the limitation of the application of a simplified and mono-factorial approach of estimating the clinical utility highlights that the assessment/testing of PGx variability should be regarded as a clinical factor contributing to the full clinical assessment. 

### 3.2. Strengths and Limitations

The population-based approach using the national health registries of a tax-financed health care system providing a free and equal health care service for everyone in Denmark, the relatively large sample size, the consistent and unambiguous data linkage of multiple registers, and the limited risk of selection bias are strengths of this study. The genotyping data with uniform quality control for the complete sample provides a solid foundation for unbiased phenotyping. Due to the data linkage, we were able to account for other independent factors and confounders that are related to drug response, including age, sex, co-medication, some somatic diseases, and pheno-conversion. We have accounted for the age differences in depression treatment and have included only the first-time use of the antidepressants of interest, which was possible due to the longitudinal design and prescription data availability going back to 1995. 

Our study also has some limitations. First, we only had information on prescriptions that were redeemed at community pharmacies in order to identify antidepressant drug users, thus, any individuals who were solely treated with antidepressants at psychiatric hospitals were not included. Moreover, we do not know if the patients actually adhered to the treatment regimen as prescribed or if they discontinued the treatment during the prescription supply. Second, we focused on people with a life-time hospital diagnosis of depression from a psychiatric hospital based on the iPSYCH study design, therefore, people using antidepressants who were solely seen by their general practitioners or by private psychiatrists were not included. However, because antidepressants can also be used for other indications than depression, e.g., anxiety or other mental or neurological disorders, the focus on people with a depression diagnosis from a psychiatric hospital makes it more likely that antidepressants were actually used for the indication of depression of similar severity. Due to the case design of the iPSYCH sample, the index drug could have been prescribed before, during, or after the registered hospital-based depression diagnosis, with 50% of the study population having had a hospital contact due to depression at the time of their first antidepressant prescription redemption. Third, the data on dosage and drug-plasma-concentrations was not available in order to evaluate the clinical significance of the genetic variations in the drug-metabolizing enzymes on the drug metabolism and the drug-plasma-concentrations as intermediates for the investigated outcomes [4]. Fourth, only four out of the eight PGx relevant SNPs for the CYP2D6 gene were available in the genotyped iPSYCH sample, with the missing variants only accounting for a summed MAFs of 0.04 [23]. Fifth, due to the missing information on the functional duplications (CYP2D6**1xN* and CYP2D6**2xN*), the CYP2D6 UM phenotypes could not be determined, but the prevalence of these duplications is only 0.8% in the Danish population [30]. Sixth, we did not account for combinatorial pharmacogenetics between the CYP2D6 and CYP2C19 genetic variants [31]. Lastly, we analyzed escitalopram and citalopram in one group, although pharmacokinetic and pharmacodynamics differences exist [32].

## 4. Methods 

### 4.1. Study Design and Setting

In this population- and register-based cohort study in Denmark, we investigated the one-year risks of developing clinical outcomes according to CYP2C19 or CYP2D6 genotypes/phenotypes in individuals with depression who had redeemed prescriptions for (es)citalopram, sertraline, or fluoxetine for the first time between 1 January 1996 and 31 December 2016. 

### 4.2. Data Sources

We used data of the Integrative Psychiatric Research (iPSYCH) consortium, which has established a large Danish population-based case-cohort sample (iPSYCH2012) [33]. Details of iPSYCH2012 have been previously described [18]. In brief, the iPSYCH sample was selected from all individuals born as singletons between 1 May 1981 and 31 December 2005 who were alive and living in Denmark at their first birthday. iPSYCH2012 included (1) a randomly selected population-based cohort of 30,000 individuals, representative of the entire Danish population born between 1981 and 2005, and (2) all individuals (cases, *n* = 57,377) who had one or more hospital-based diagnoses of five selected severe mental disorders by 31 December 2012, including schizophrenia, affective disorders, bipolar disorder, attention-deficit/hyperactivity disorder (ADHD), and autism spectrum disorder [18]. 

iPSYCH2012 is linked via the anonymized personal identification number assigned to the residents of Denmark at birth or immigration to longitudinal data of the following: (i) The Danish Civil Registration System [34]; (ii) The Danish Psychiatric Central Research Register (PCRR) [35]; (iii) The Danish National Prescription Registry [36] holding information on prescriptions redeemed at community pharmacies since 1995; (iv) the Danish National Patient Register [37]; (v) the Danish Register of Causes of Death [38]; (vi) the socio-demographic and labor market-related data hosted at Statistics Denmark (DST) [39]; and (vii) the Danish Neonatal Screening Biobank [40], which stores dried blood spots from practically all neonates in Denmark. 

### 4.3. Genotyping and Phenotyping

Genetic information of the individuals in iPSYCH2012 was collected from the dried blood spots retrieved from the Danish Neonatal Screening Biobank [40]. A total of 80,422 samples were genotyped using the Infinium PsychChip v1.0 array (Illumina, San Diego, CA, USA). The array-based genotyped SNPs were imputed using 1000 Genomes Project phase 3, with GRCh37 as a reference. [41]. After sample and genotype QC using the Ricopilli bioinformatics pipeline [42], 6,361,597 high-quality best guess-genotypes were available. See Schorck et al. 2019 for a detailed description of the imputation and QC procedures [43].

### 4.4. Study Population and Study Period

From iPSYCH2012, we identified all individuals with a hospital-based depression diagnosis of F32–33 according to the International Classification of Diseases, 10th edition (ICD-10) as psychiatric inpatient, outpatient, or emergency department admissions at any time between 1 January 1996 and 31 December 2012 [44]. Of those. we included all individuals redeeming a first-time (i.e., incident new-use since 1995) prescription for (es)citalopram, sertraline, or fluoxetine between 1 January 1996 and 31 December 2016 (study period, Figure 2) [45]. Thus, we did not include individuals diagnosed with a mental disorder before the start of the study period (1 January 1996) or a prescription redemption for an SSRI of interest in 1995. We defined the date of the first prescription redemption of the respective drug as the index date. 

### 4.5. CYP2C19 and CYP2D6 Genotyping and Phenotyping

The exposure was defined as expression of any mutant CYP2C19 (UM, RM, IM, or PM) or CYP2D6 genotype/phenotype (IM or PM) based on the array-based SNP information [46,47]. Non-exposed individuals who did not carry a mutant CYP2C19 or CYP2D6 genetic variant and were classified as NM. The PGx phenotype translation procedure has been described in detail [46]. In short, the translation based on the 2019 Ubiquitous-PGx (U-PGx) panel [23] included 9 variants for CYP2C19 and 8 variants for CYP2D6, including CYP2D6 duplication and deletion. The SNPs were linked to star (*) allele nomenclature, which standardizes genetic polymorphism annotations for cytochrome P450 genes to simplify the translations of a patient’s genotype into a predicted clinical phenotype [48,49,50]. For CYPC19, the star-alleles **2*, **8*, and **17* variants and the CYPCD6 the star-alleles **4*, **10*, **17*, and **41* were available. Based on the individuals’ genotypes, the diplotypes were translated into the PGx phenotypes (Appendix A). 

### 4.6. Outcomes 

Outcomes were assessed within one year of the index date (Figure 1) as follows: (1) Switching from the index prescription to any other antidepressant (ATC: N06A); (2) Discontinuation, defined as less than three prescriptions of the index antidepressant; (3) Emergency department contact at a psychiatric hospital; (4) Suicide attempt/self-harm, which was identified according to the algorithm as described by Gasse et al. [51].

### 4.7. Covariates

Detailed definitions of all the covariates and confounders are shown in Appendix A. Covariates included the following: age, gender, region of index prescription, socio-economic status (SES, assessed for the adults of the study population and for the parents of the children of the study population), number of previous psychiatric diagnoses, prescription drug use acting as CYP2C19/CYP2D6 inhibitor/inducer within the last three months of the index date, calendar year of index prescription, any hospital contacts within the previous year of index date, previous suicide attempt/self-harm, and antiepileptic drug use within the last three months prior to the index date. 

### 4.8. Statistical Analyses

The analysis was performed separately for the three age groups (children and adolescents, young adults, adults) because of the differences in disease states, enzyme activity, experience of adverse events, and choice of antidepressant treatment in children and adults. Children and adolescents with depression must be referred to a pediatric psychiatrist before treatment initiation, irrespective of the severity of depression, according to Danish guidelines [13]. Young adults with depression should be referred to a psychiatrist within one week after the start of antidepressant treatment. Adults are often treated solely by general practitioners and are only recommended to be referred to a psychiatric department if treatment with two different antidepressants has failed or suicidality is suspected, in diagnostic doubts, and/or the presence of psychotic symptoms or somatic disorders that complicate treatment with antidepressants [52].

We described the characteristics of the study population at the index date as proportions (%). We followed all individuals from the index date for one year. Individuals were censored at end of follow-up, outcome events, emigration, or death, whichever came first. We calculated incidence rates (IR) using SAS %Lexis macro [53]. The IRs were modeled in a Poisson regression analysis (using follow-up as offset) to investigate the association between CYP2C19/CYP2D6 phenotype status and clinical outcomes, which was presented as incidence rate ratios (IRR) with 95% confidence intervals (CI). We considered a 95% CI that did not overlap 1.00 to be statistically significant.

All analyses were adjusted for the following potential covariates and confounders: age, gender, region of index prescription, socio-economic status (SES, assessed for the adults of the study population and for the parents of the children of the study population (Appendix A), number of previous psychiatric diagnoses, CYP2C19/CYP2D6 inhibitor/inducer use within the last three months of the index date, and calendar year of index prescription. For emergency department contact, we further adjusted for any hospital contacts within the previous year of the index date. For the outcome of suicide attempt/self-harm, we additionally adjusted for previous suicide attempt/self-harm and for antiepileptic drug use within the last three months of index date. 

We have reported the potential clinical utility of PGx testing for CYP2C19 and CYP2D6 genetic variability by calculating the population attributable fraction (PAF), number needed to treat (NNT), and number needed to genotype (NNG) for all significant associations, based on Tonk et al. [16]. 

All data processing and analyses were carried out using SAS statistical software version 9.4 (SAS Institute Inc., Cary, NC, USA). 

### 4.9. Data Protection

Data permissions have been granted to iPSYCH2012 by The Danish Scientific Ethics Committee (EC: 1-10-72-287-12), the Danish Health Data Authority, the Danish data protection agency, and the Danish Neonatal Screening Biobank Steering Committee. Danish Data Protection Agency: Journal number 2015-57-0002 /Journal number: 62908 (Umbrella permission Aarhus University) and National Board of Health: FSEID 00000098. Researchers can access anonymous individual-level data only through secure servers where the download of individual-level information is prohibited, which protects the privacy of the individuals included in the study. Due to data protection, we do not report numbers below five, but state ‘<5’, or combine categories to achieve larger counts than five.

## 5. Conclusions

Our study adds new knowledge of the associations between CYP2C19 and CYP2D6 phenotypes and antidepressant switching, discontinuation, emergency department contacts, and suicide attempt/self-harm in children, adolescents, and adults with depression with incident new-use of (es)citalopram, sertraline, or fluoxetine, which indicates the clinical utility of PGx in patients with depression. Even though the associations were strong and pronounced from a population perspective, the nominal clinical utility remains low, due to the multifactorial contribution of many factors to the outcomes. Children and adolescents seem to be a relevant target group benefitting from PGx testing where clinical data is still rare and urgently needed, because a large part of the existing evidence is deduced from adult data.

## Figures and Tables

**Figure 1 pharmaceuticals-15-00870-f001:**
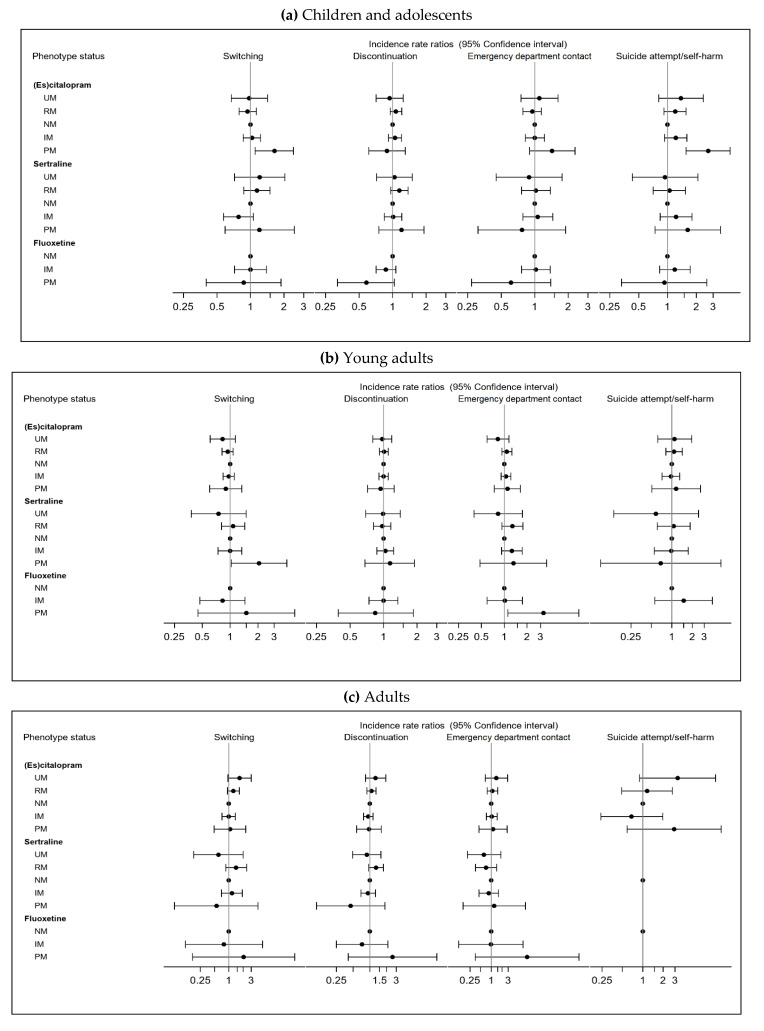
Adjusted incidence rate ratios (IRR) and 95% confidence intervals of the associations between the CYP2C19 and CYP2D6 phenotypes and clinical outcomes in people with a hospital depression diagnosis between 1 January 1996 and 31 December 2012 and a first-time prescription for (es)citalopram, sertraline, or fluoxetine between 1 January 1996 and 31 December 2016, stratified by age groups (≤18, 19–25, 26+ years). Abbreviations: UM: ultra-rapid metabolizer, RM: rapid metabolizer, NM: normal metabolizer, IM: intermediate metabolizer, PM: poor metabolizer. NM was the reference group. IRR were adjusted for: age, gender, region of index prescription, socio-economic status (SES), number of previous psychiatric diagnosis, CYP2C19/CYP2D6 inhibitor and inducer use within the last three months of index date, and calendar year of index prescription. For emergency department contact, we further adjusted for any hospital contacts within the previous year of index date. For the outcome of suicide attempt/self-harm, we also adjusted for previous suicide attempt/self-harm and for antiepileptic drug use within the last three months of index date.

**Figure 2 pharmaceuticals-15-00870-f002:**
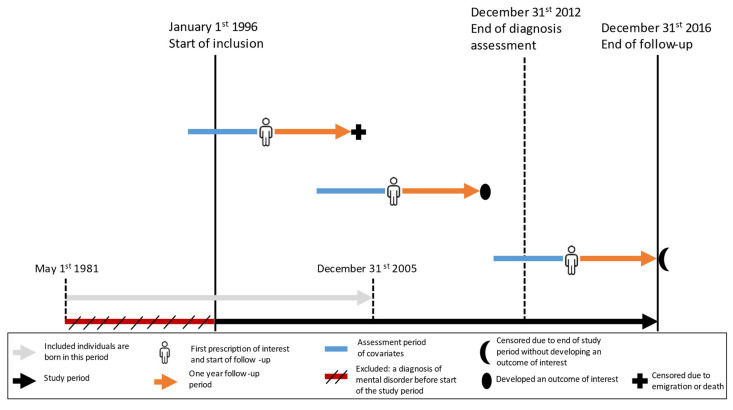
The selection of the study population, study design, and follow-up.

**Table 1 pharmaceuticals-15-00870-t001:** Baseline characteristics at the index date of the first-time prescription of sertraline, escitalopram, citalopram, and fluoxetine of the total study population (*n* = 17,297) of all individuals born between 1981 and 2005 with a depression diagnosis any time between 1996 and 2012.

	Antidepressants							
	Citalopram, *n* = 8281		Escitalopram, *n* = 2632		Sertraline, *n* = 4583		Fluoxetine, *n* = 1801	
	*n*	(%)	*n*	(%)	*n*	(%)	*n*	(%)
**Sex**								
Female	5896	(71.2)	1783	(67.7)	3164	(69.0)	1377	(76.5)
Male	2385	(28.8)	849	(32.3)	1419	(31.0)	424	(23.5)
**Age in groups**								
Children/adolescents (≤18 years)	3111	(37.6)	928	(35.3)	2513	(54.8)	1338	(74.3)
Young adults (19–25 years)	4263	(51.5)	1428	(54.3)	1567	(34.2)	403	(22.4)
Adults (26+ years)	907	(11.0)	276	(10.5)	503	(11.0)	60	(3.3)
**Mean age in years, (SD)**	20.3 (3.6)		20.5 (3.4)		19.2 (4.3)		17.5 (3.3)	
**Region at index prescription**								
Capital Region	2296	(27.7)	787	(29.9)	1068	(23.3)	514	(28.5)
Middle Jutland	2029	(24.5)	676	(25.7)	1161	(25.3)	323	(17.9)
North Jutland	887	(10.7)	227	(8.6)	574	(12.5)	148	(8.2)
Southern Denmark	1592	(19.2)	595	(22.6)	1073	(23.4)	374	(20.8)
Zealand	1477	(17.8)	347	(13.2)	707	(15.4)	442	(24.5)
**Parents/adults SES ***								
Missing	152	(1.8)	23	(0.9)	126	(2.7)	25	(1.4)
Employed	3670	(44.3)	1256	(47.7)	2292	(50.0)	1094	(60.7)
On social benefits	1839	(22.2)	465	(17.7)	1064	(23.2)	309	(17.2)
On study	1947	(23.5)	673	(25.6)	782	(17.1)	273	(15.2)
Others	673	(8.1)	215	(8.2)	319	(7.0)	100	(5.6)
**Within last year: No. of psychiatric hospital contacts**								
0	4965	(60.0)	1450	(55.1)	2078	(45.3)	590	(32.8)
1	1220	(14.7)	408	(15.5)	844	(18.4)	371	(20.6)
2	445	(5.4)	143	(5.4)	311	(6.8)	193	(10.7)
3	213	(2.6)	80	(3.0)	143	(3.1)	68	(3.8)
4	107	(1.3)	36	(1.4)	100	(2.2)	42	(2.3)
>4	1331	(16.1)	515	(19.6)	1107	(24.2)	537	(29.8)
**Past: No. of past mental diagnoses**								
0	2757	(33.3)	798	(30.3)	952	(20.8)	241	(13.4)
1	2511	(30.3)	880	(33.4)	1217	(26.6)	566	(31.4)
2	1648	(19.9)	550	(20.9)	1169	(25.5)	603	(33.5)
3	845	(10.2)	268	(10.2)	779	(17.0)	253	(14.0)
4	361	(4.4)	105	(4.0)	296	(6.5)	96	(5.3)
>4	159	(1.9)	31	(1.2)	170	(3.7)	42	(2.3)
**Past ever: history of self-harm/suicide attempt**								
Yes	1312	(15.8)	431	(16.4)	651	(14.2)	321	(17.8)
**Within last year: history of self-harm/suicide attempt**								
Yes	586	(7.1)	206	(7.8)	346	(7.5)	197	(10.9)
**Within last 90ds: strong CYP2D6 inhibitor use**								
Yes	67	(0.8)	32	(1.2)	66	(1.4)	10	(0.6)
**Within last 90ds: moderate CYP2D6 inhibitor use**								
Yes	142	(1.7)	42	(1.6)	77	(1.7)	14	(0.8)
**Within last 90ds: weak CYP2D6 inhibitor use**								
Yes	One of the categories had <5 observations							
**Within last 90ds: strong CYP2C19 inhibitor use**								
Yes	216	(2.6)	73	(2.8)	109	(2.4)	39	(2.2)
**Within last 90ds: moderate CYP2C19 inhibitor use**								
No	8281	(100.0)	2632	(100.0)	4583	(100.0)	1801	(100.0)
**Within last 90ds: weak CYP2C19 inhibitor use**								
Yes	189	(2.3)	59	(2.2)	122	(2.7)	41	(2.3)
**Within last 90ds: CYP2C19 inducer use**								
Yes	All categories had <5							
**Within last 90ds: Antiepileptic drug use**								
Yes	80	(1.0)	35	(1.3)	69	(1.5)	13	(0.7)
**Year as category of first prescription**								
1995–2001	237	(2.9)	0	0	185	(4.0)	44	(2.4)
2001–2005	2201	(26.6)	524	(19.9)	988	(21.6)	264	(14.7)
2006–2010	4447	(53.7)	1890	(71.8)	1931	(42.1)	828	(46.0)
2011–2016	1396	(16.9)	218	(8.3)	1479	(32.3)	665	(36.9)

* For those who had missing information on their own socioeconomic status (SES) we extracted SES from their parents. For a detailed description of all the variables see Appendix A. Ds = days.

**Table 2 pharmaceuticals-15-00870-t002:** Prevalence of CYP2C19 and CYP2D6 phenotypes of individuals born between 1981 and 2005, with a depression diagnosis any time between 1996 and 2012, with at least one prescription for escitalopram, citalopram, sertraline, or fluoxetine.

	Antidepressants									
	Total, *n* = 17,297		Escitalopram, *n* = 2632		Citalopram, *n* = 8281		Sertraline, *n* = 4583		Fluoxetine, *n* = 1801	
	*n*	(%)	*n*	(%)	*n*	(%)	*n*	(%)	*n*	(%)
CYP2D6 phenotype										
CYP2D6_NM	10,770	(62.3)	1629	(61.9)	5159	(62.3)	2855	(62.3)	1127	(62.6)
CYP2D6_IM	5781	(33.4)	873	(33.2)	2778	(33.5)	1533	(33.4)	597	(33.1)
CYP2D6_PM	746	(4.3)	130	(4.9)	344	(4.2)	195	(4.3)	77	(4.3)
CYP2C19 phenotype										
CYP2C19_UM	678	(3.9)	118	(4.5)	304	(3.7)	194	(4.2)	62	(3.4)
CYP2C19_RM	4483	(25.9)	687	(26.1)	2168	(26.2)	1143	(24.9)	485	(26.9)
CYP2C19_NM	7553	(43.7)	1122	(42.6)	3600	(43.5)	2042	(44.6)	789	(43.8)
CYP2C19_IM	4215	(24.4)	652	(24.8)	2024	(24.4)	1111	(24.2)	428	(23.8)
CYP2C19_PM	368	(2.1)	53	(2)	185	(2.2)	93	(2)	37	(2.1)

Abbreviations: NM: normal metabolizer, IM: intermediate metabolizer, PM: poor metabolizer, RM: rapid metabolizer, UM: ultrarapid metabolizer.

**Table 3 pharmaceuticals-15-00870-t003:** Measures of population impact of pharmacogenetic testing.

	**Age Group**	**Children and Adolescents**	**Children and Adolescents**	**Young Adults**	**Young Adults**
	Drug	(Es)citalopram	(Es)citalopram	Sertraline	Fluoxetine
	Phenotype	CYP2C19 PM	CYP2C19 PM	CYP2C19 PM	CYP2D6 PM
	Risk geno-/phenotype freq.	2.18%	2.18%	2.00%	4.30%
	**Outcome**	**Switching**	**Suicide Attempt/Self-Harm**	**Switching**	**ER Contact**
	IRR *	1.64	2.67	2.06	3.28
	RR	1.46	2.15	1.57	**
	RD	0.09	0.1	0.11	**
**Population impact of PGx**	PAF	1.00%	2.4%	1.12%	**
	NNT	11	11	10	**
	NNG	503	464	460	**

Abbreviations: PM: poor metabolizer, IRR: incidence rate ratio, RR: relative risk, RD: risk difference, PAF: population attributable fraction, NNT: number needed to treat, NNG: number needed to genotype. * Adjusted for: age, gender, region of index prescription, socio-economic status (SES), number of previous psychiatric diagnosis, CYP2C19/CYP2D6 inhibitor and inducer use within the last three months of index date, and calendar year of index prescription. For emergency room contact, we further adjusted for any hospital contacts within the previous year of index date. For the outcome of suicide attempt/self-harm, we also adjusted for previous suicide attempt/self-harm and for antiepileptic drug use within the last three months of index date. ** Numbers cannot be calculated because there were <5 cases for at least one of the needed numbers. See Appendix A for the underlying numbers for the calculation.

## Data Availability

Data is contained within the article and Appendix A.

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
