# Peer review of "Clinical Impact of Functional CYP2C19 and CYP2D6 Gene Variants on Treatment with Antidepressants in Young People with Depression: A Danish Cohort Study"

_pharmaceuticals, 2022, doi:10.3390/ph15070870_

Round 1

Reviewer 1 Report

The present study represents a very impressive sample size. The authors studied more than 17,000 people, adults and children.

This study is of great value also because of its coverage of age-related aspects. It is important to demonstrate the differences between children and adults, which affect pharmacogenetics among other things, on the basis of native material.

The comments I could have made are already contained in the Limitations section. The authors have detailed what the shortcomings of their work are. But it still remains unclear why it was not possible to analyse the combined effect of CYP2D6 and CYP2C19 phenotypes? Fluoxetine is known to be metabolised by both isoenzymes. This is a likely reason why no pharmacogenetic testing recommendations for prescribing fluoxetine have yet been developed. I assume that the authors of this article have data on the CYP2C19 and CYP2D6 phenotype at their command. In that case it seems to me that an evaluation of the effects of both isoenzymes on the safety of fluoxetine would be a very valuable part of this study.

Of course, if the authors have difficulty refining this section, this is not an obstacle to accepting the article for publication. But I would recommend authors to explain exactly in the text why they have refused to do this analysis.

Should the authors plan to analyse the combined effects of CYP2C19 and CYP2D6 on the safety of fluoxetine in children, this would be very interesting material for a new article.

Author Response

Reviewer 1

The present study represents a very impressive sample size. The authors studied more than 17,000 people, adults and children.

This study is of great value also because of its coverage of age-related aspects. It is important to demonstrate the differences between children and adults, which affect pharmacogenetics among other things, on the basis of native material.

The comments I could have made are already contained in the Limitations section. The authors have detailed what the shortcomings of their work are.

But it still remains unclear why it was not possible to analyse the combined effect of CYP2D6 and CYP2C19 phenotypes? Fluoxetine is known to be metabolised by both isoenzymes. This is a likely reason why no pharmacogenetic testing recommendations for prescribing fluoxetine have yet been developed. I assume that the authors of this article have data on the CYP2C19 and CYP2D6 phenotype at their command. In that case it seems to me that an evaluation of the effects of both isoenzymes on the safety of fluoxetine would be a very valuable part of this study.

Of course, if the authors have difficulty refining this section, this is not an obstacle to accepting the article for publication. But I would recommend authors to explain exactly in the text why they have refused to do this analysis.

Should the authors plan to analyse the combined effects of CYP2C19 and CYP2D6 on the safety of fluoxetine in children, this would be very interesting material for a new article.

RESPONSE:

We would like to thank Reviewer 1 for the positive evaluation.

It is correct that we had both CYP2D6 and CYP2C19 phenotypes at our command.

We focused on CYP2D6 as a potential relevant, yet understudied drug gene interaction with fluoxetine as it has been previously evaluated by the Royal Dutch Pharmacists Association - Pharmacogenetics Working Group  (https://www.pharmgkb.org/chemical/PA449673/guidelineAnnotation/PA166182852).

We agree with the reviewer that the consideration of both the enzymes would be valuable, but given the complexity of the metabolizing pathway as also discussed in the current paper (Page 13) and https://www.pharmgkb.org/pathway/PA161749012, it would have been beyond the scope of the current study. We also face power issues considering the different combinations of possible phenotypes of both the enzymes in relation to outcomes. We definitely agree that it would be interesting material for a new article choosing, in a larger data set, a combinatorial approach, e.g. based on activity scores as suggested by van Westrhenen et al. (van Westrhenen R, Aitchison KJ, Ingelman-Sundberg M, Jukić MM. Pharmacogenomics of Antidepressant and Antipsychotic Treatment: How Far Have We Got and Where Are We Going? Front Psychiatry. 2020 Mar 12;11:94. doi: 10.3389/fpsyt.2020.00094. PMID: 32226396; PMCID: PMC7080976.)

To accommodate the reviewer's comment, we have reorganized the related paragraph in the discussion (in green) and added a clarification (in yellow) (Page 13):

Regarding fluoxetine, according to previous reports, CYP2D6 metabolizer status showed no influence on 8- or 12-week fluoxetine treatment response, assessed with multiple disease severity scales, in children and adolescents [48]. Here, we found nominally decreased risks for all outcomes in children and adolescents with CYP2D6 PM status, which may indicate a superior response to fluoxetine in CYP2D6 PMs, possibly due to higher drug-plasma concentrations in these patients [4], without the off-set of higher risks of adverse events leading to discontinuation or switching. In contrast to children and adolescents, young adults and adults using fluoxetine with PM status in our study had an increased risk of switching and emergency room contacts, which is in line with a smaller study reporting that 33% with a CYP2D6 PM status discontinued fluoxetine treatments compared with 14% of adults with a CYP2D6 NM status [49] The metabolism of fluoxetine is complicated by self-inhibition/pheno-conversion of CYP2D6 by fluoxetine enantiomers during chronic treatment, which increases the importance of alternative metabolic pathways including CYP2C19 [13]. Thus CYP2C19 metabolism and other alternative pathways may compensate limited CYP2D6 metabolism [50]. Due to the described complexity of the metabolic pathway and power issues regarding treatment outcomes, it would have been beyond the scope of the current study to evaluate the combinatorial effect of both the CYP2D6 and CYP2C19 variants, which should be addressed in future studies.

Reviewer 2 Report

I would like to congratulate the authors on this large cohort study investigating the role of CYP2C19 and CYP2D6 genetic variants in the treatment of depression with SSRIs. Working within the limitations of their cohort, the authors have done an awesome job in teasing out the effects reported in this manuscript. Even though, the results may not have a massive impact on prescribing, this study will add to the current knowledge in this area. 

I do not have any major concerns with this manuscript. 
However, there are a couple of minor points that I would like to raise to the attention of the authors. 

1. Page 4 under 2.5: authors have used the word 'deviant' twice in this paragraph. I would suggest changing 'deviant' to 'mutant'.

2. Permitting the requirements of this journal, it might be better to split table 1 into two tables for the ease of reading the information presented in table 1.

3. Page 14, ilmitations: authors have used the word 'ingested'. I would recommend changing this to 'adhered to treatment regimen as prescribed'. 

4.  Would the authors add a line or two to explain how not having dosage and drug-plasma-concentrations is a limitation? This would help in the understanding of younger researchers not familiar with pharmacogenomics studies. 

Author Response

We would like to thank Reviewer 2 for the positive evaluation and the constructive comments, which we have addressed point by point.

  1. Page 4 under 2.5: authors have used the word 'deviant' twice in this paragraph. I would suggest changing 'deviant' to 'mutant'.

RESPONSE 1: We changed the wording to 'mutant' in both cases.

  1. Permitting the requirements of this journal, it might be better to split table 1 into two tables for the ease of reading the information presented in table 1.

RESPONSE 2: We adapted the table in that we changed the sequence of the columns in line with the other tables. However, the online version allows us to show the complete table with scrolling function and formatting will be taken care of by the editorial office.  

  1. Page 14, ilmitations: authors have used the word 'ingested'. I would recommend changing this to 'adhered to treatment regimen as prescribed'. 

RESPONSE 3: We changed the wording according to reviewer's suggestion.

  1. Would the authors add a line or two to explain how not having dosage and drug-plasma-concentrations is a limitation? This would help in the understanding of younger researchers not familiar with pharmacogenomics studies. 

RESPONSE 4: Thank you for the suggestions. We added (yellow):

"Third, data on dosage and drug-plasma-concentrations was not available to evaluate the clinical significance of the genetic variations in the drug metabolizing enzymes on drug metabolism and drug-plasma-concentrations as intermediates for the investigated outcomes [4]."